# Photo-cycloaddition reactions of vinyldiazo compounds

Ming Bao[1], Klaudia Łuczak [2], Wojciech Chaładaj [2] ✉, Marriah Baird[1], Dorota Gryko [2] ✉ & Michael P. Doyle [1] ✉

Heterocyclic rings are important structural scaffolds encountered in both natural and synthetic compounds, and their biological activity often depends on these motifs. They are predominantly accessible via cycloaddition reactions, realized by either thermal, photochemical, or catalytic means. Various starting materials are utilized for this purpose, and, among them, diazo compounds are often encountered, especially vinyldiazo compounds that give access to donor-acceptor cyclopropenes which engage in [2+$n$] cycloaddition reactions. Herein, we describe the development of photochemical processes that produce diverse heterocyclic scaffolds from multisubstituted oximido-vinyldiazo compounds. High chemoselectivity, good functional group tolerance, and excellent scalability characterize this methodology, thus predisposing it for broader applications. Experimental and computational studies reveal that under light irradiation these diazo reagents selectively transform into cyclopropenes which engage in cycloaddition reactions with various dipoles, while under thermal conditions the formation of pyrazole from vinyldiazo compounds is favored.

The development of effective, sustainable methodologies giving access to valuable heterocyclic molecules is of paramount importance to modern synthetic chemistry. Among approaches to their synthesis, cycloaddition reactions stand at the forefront, and the use of visible light as the only source of energy for these transformations is highly appealing. Recently, diazo compounds that produce various reactive intermediates have attracted considerable interest[1–7]. In particular, vinyldiazo compounds have proven to be valued reactants for the synthesis of diverse carbo- and heterocyclic compounds[8–10], especially through cycloaddition reactions[11,12]. Their catalytic applications include highly enantioselective [3 + 3]-, [3 + 2]-, and [3 + 1]-cycloaddition[13]. Furthermore, silyl-protected enol diazoacetates are known to form stable donor-acceptor cyclopropenes that provide a resting state for incipient metallovinyl carbenes in cycloaddition reactions[14]. Their photochemically induced dinitrogen extrusion to form vinylcarbene intermediates has been investigated[15,16], and the resultant formation of cyclopropene products with alkyl and aryl substituents is well known[17–19]. Cyclopropenes are highly strained

alkenes that are, themselves, highly reactive in cycloaddition reactions[20,21]; they are activated by strain to undergo "click"-like cycloaddition reactions and although few reports have documented these transformations[13,22–25], they have already been employed for site-specific protein conjugation[26,27], cell labeling[28,29] and other materials development functions[30,31].

The linkage between vinyldiazo compounds and cyclopropenes provides versatility in product formation. Donor-acceptor cyclopropenes are produced catalytically from silyl-protected enoldiazo compounds via metallovinyl carbene intermediates[32], as well as thermally[33] in nearly quantitative yield (Fig. 1A). β-Aryl/alkyl vinyl diazocarbonyl compounds undergo non-reversible catalytic formation of unstable cyclopropenes via metallovinyl carbenes[34], but thermally they form 3$H$-pyrazoles that transform into 1$H$-pyrazoles by 1,5-H transfer[35]. In particular, styryldiazoacetates undergo the thermal formation of 1$H$-pyrazoles, but they do not form cyclopropenes in catalytic reactions[36]. In catalytic reactions with vinyldiazo compounds, dipolar species can intercept metallo-vinylcarbenes prior to cyclopropene

[1]Department of Chemistry, The University of Texas at San Antonio, San Antonio, Texas, USA. [2]Institute of Organic Chemistry Polish Academy of Sciences, Kasprzaka 44/52, Warsaw, Poland. ✉e-mail: wojciech.chaladaj@icho.edu.pl; dorota.gryko@icho.edu.pl; michael.doyle@utsa.edu

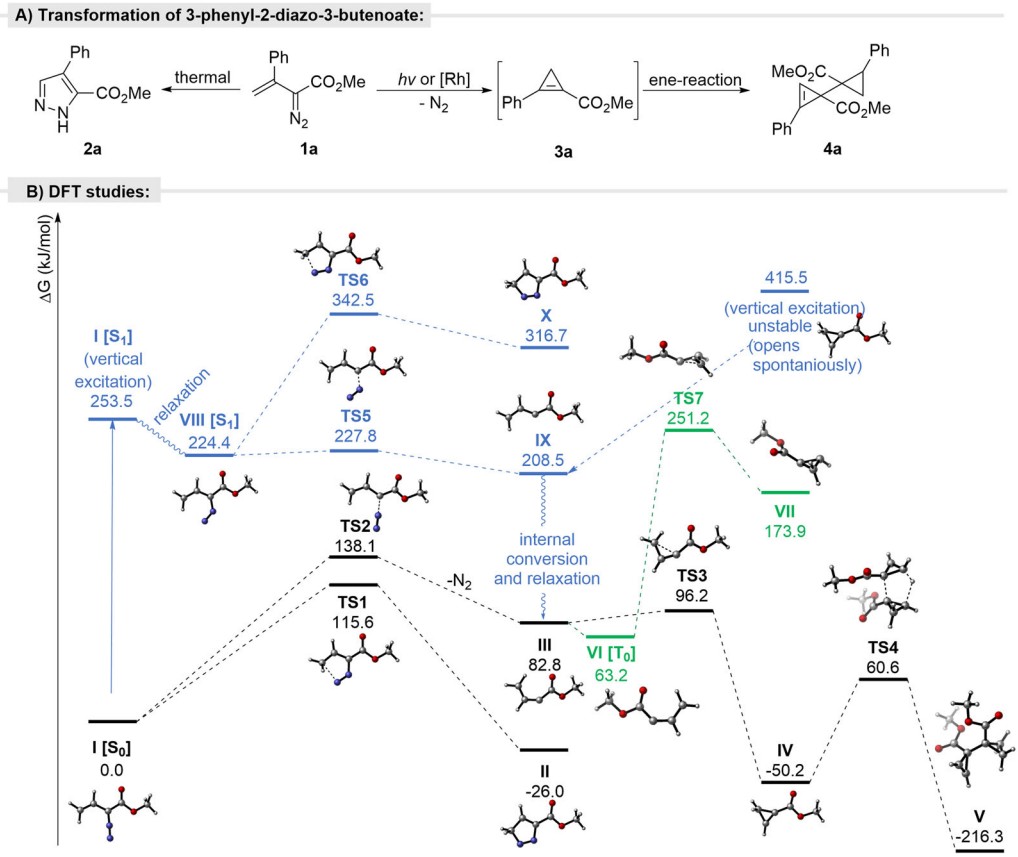

**Fig. 1 | Transformations of vinyldiazo compounds. A** Catalytic and thermal formation of cyclopropenes from vinyldiazo compounds; **B** Cyclopropene cycloaddition reactions; **C** Tandem cycloaddition reactions of vinyldiazo compounds.

**Fig. 2 | Photoreactivity of vinyl diazo compounds. A** Transformation of 3-phenyl-2-diazo butanoate. **B** Calculated reactivity of methyl 2-diazobut-3-enoate in the ground and exited states.

formation but with suitable donor-acceptor substituents; the cyclopropene can be returned to the metallo-vinylcarbene to produce [3+n]-cycloaddition products[14]. Because of their inherent reactivity, the double bond in cyclopropenes undergoes [2+n]-cycloaddition with viable dipolar species (Fig. 1B)[20,21,37,38]. However, although vinyldiazo compounds are precursors to cyclopropenes, they have not been used as direct precursors for [2+n]-cycloaddition products that can be achieved with cyclopropenes; instead, the [2+n]-cycloaddition products are formed from cyclopropenes, which, in turn, are separately prepared from vinyldiazo compounds or by alternative methods. We now report a general, efficient methodology for the conversion of vinyldiazo compounds to [2+n]-cycloaddition products through the photochemical production of intermediate cyclopropenes (Fig. 1C).

## Results and Discussion

### Generation of cyclopropenes from vinyldiazo compounds

Vinyldiazo compounds form either cyclopropenes (dinitrogen extrusion)[33,39] or 1H-pyrazoles (cycloaddition/rearrangement)[34,36]. This dichotomy is found in the photochemical and thermal reactions of β-aryl-/alkyl-vinyl diazoacetates. The thermal reaction of 3-phenyl-2-diazo-3-butenoate (**1a**) produces corresponding 1H-pyrazole **2a** in high yield, whereas dinitrogen is lost in the photochemical reaction to give cyclopropene dimer **4a** resulting from an ene-reaction[40] of the unstable cyclopropene intermediate **3a** (Fig. 2A). The same **4a** is obtained in the rhodium acetate catalyzed reaction of **1a**[34].

To understand the basis for the divergent outcomes from thermal and photochemical reactions, the chemistry of model methyl

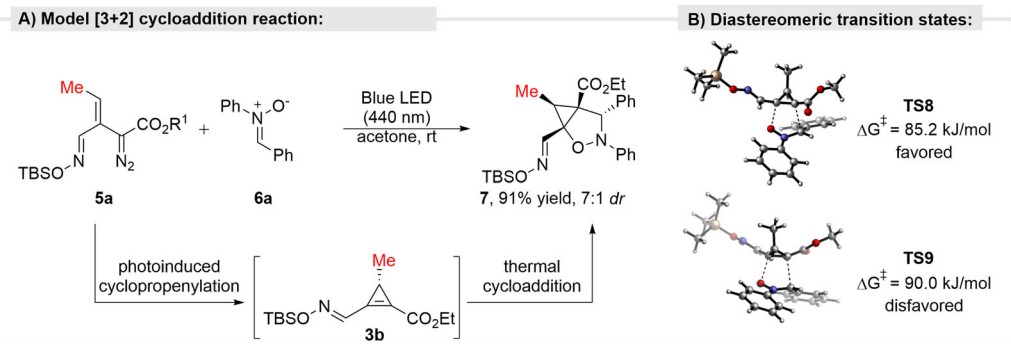

**Fig. 3 | [3 + 2]-Cycloaddition reaction of diazo compounds. A** Model reaction of oximidovinyldiazo acetate **5a** with nitrone **6. B** DFT calculations on the cycloadditon [3 + 2] of model cyclopropene with nitrone **6**.

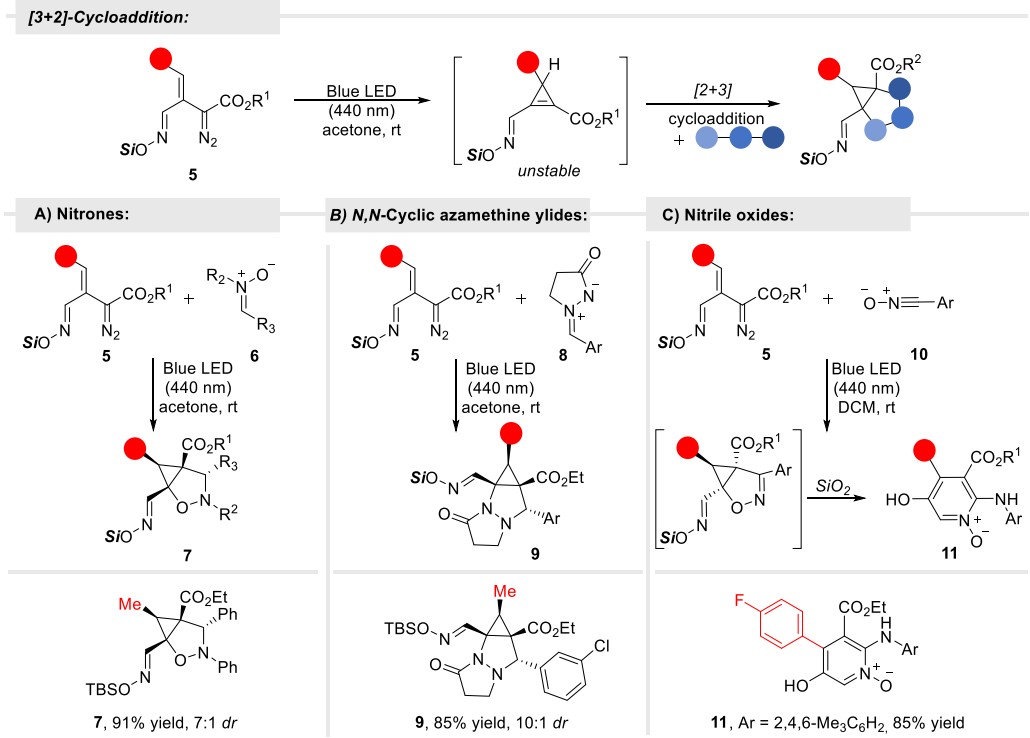

**Fig. 4 | Cycloaddition reactions of oximidovinyldiazo compounds with various dipoles. A** Model reaction of oximidovinyldiazo acetate **5** with nitrone **6. B** Model reaction of oximidovinyldiazo acetate **5** with *N,N*-cyclic azamethine ylide **8. C** Model reaction of oximidovinyldiazo acetate **5** with nitrile oxide **10**.

2-diazobut-3-enoate **I** was investigated computationally on S0 and S1 potential energy surfaces (Fig. 2B)[41,42]. In the ground state (S0 PES), cyclization of the model vinyl diazoacetate **I** towards pyrazole **II** is preferred by 22.5 kJ/mol over extrusion of nitrogen ($\Delta G^{\ddagger}$ = 115.6 and 138.1 kJ/mol, respectively, Fig. 2B), although both are hardly accessible at ambient temperature. The latter of the discussed pathways provides carbene **III**, which easily ($\Delta G^{\ddagger}$ = 13.4 kJ/mol) cyclizes, yielding cyclopropane **IV**. The dimerization of **IV** via an ene-reaction must overcome a barrier of $\Delta G^{\ddagger}$ = 110.8 kJ/mol. A similar reactivity pattern was observed, i.e., preference for cyclization over extrusion of $N_2$ under thermal conditions, for vinyl diazo compounds substituted with simple alkyl and aryl groups (see, SI). Conversely, the calculated mechanistic scenario on the S1 PES is considerably different. The cycloisomerization of methyl 2-diazobut-3-enoate **VIII** to pyrazole system **X** not only features a considerable barrier ($\Delta G^{\ddagger}$ = 118.1 kJ/mol) but is also highly endergonic ($\Delta G$ = 92.3 kJ/mol). In contrast, the excited vinyl diazoacetate **VIII** loses dinitrogen in a practically barrierless event ($\Delta G^{\ddagger}$ = 3.4 kJ/mol). However, the subsequent cyclization of

resulting carbene **IX** into a cyclopropene ring cannot occur at S1 PES (high-energy cyclopropene in the excited state spontaneously opens towards the vinyl carbene). Thus, it presumably first undergoes internal conversion to ground state **III**, at which it then converts into cyclopropene system **IV**. In contrast, cyclization of carbene **VI** in the triplet state, which is a ground state[15,16], through **TS5** is associated with a very high barrier of $\Delta G^{\ddagger}$ = 188.0 kJ/mol, which is not accessible under the reaction conditions.

## Cycloaddition reactions of vinyldiazo compounds

Based on these theoretical studies, we expected that cyclopropenes could be generated from vinyldiazo compounds under light irradiation and, as such, engage in reactions with dipoles. In this regard, we previously observed traces of [3 + 2]-cycloaddition product formed via cyclopropene intermediates, accompanying the intended metallovinylcarbene [3+n]-cycloaddition product, in the Rh-catalyzed reaction of enol diazoacetates with isoquinolinium dicyanomethylides[32,34]. To evaluate the possibility of developing photochemical tandem reaction,

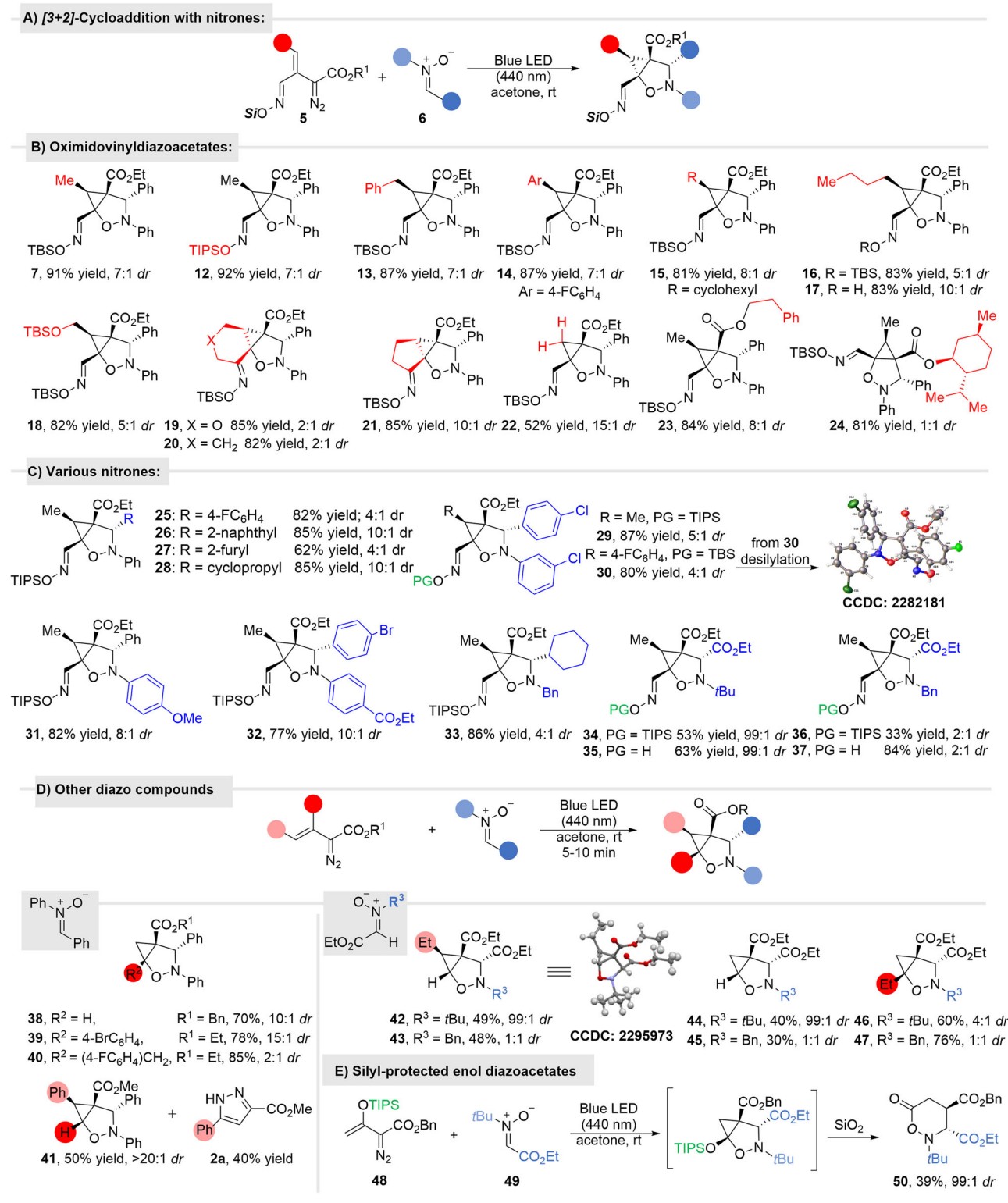

**Fig. 5 | Photocatalytic formal [3 + 2]-cycloaddition of diverse oximidovinyldiazo esters with nitrones. A** The general transformation; **B** Reaction scope of oximidovinyldiazoacetates; **C** Reaction scope of nitrones; **D** Reaction of other types of vinyldiazo compounds with nitrones; **E** Reaction of silyl-protected enol diazoacetate **48**.

generation of the cyclopropene from vinyldiazo compounds followed by its reaction with dipoles, leading to heterocycles, oximidovinyldiazo acetate **5a** was chosen as a model diazo compound (Fig. 3). Such vinyldiazo reagents are conveniently prepared from 1,2,3-triazine 1-oxides[43]. Since cyclopropene cycloaddition with nitrones is known to occur in reactions involving metallo-vinylcarbene intermediates[44–48],

we commenced our studies with nitrones **6a** as a reliable model dipole[11,49]. Photolysis with blue light gave the expected isooxazolidine cycloaddition product **7** as a mixture of diastereoisomers (7:1 ratio) in 91% yield. Control experiments revealed that the reaction is photochemical in nature. Under light irradiation cyclopropene forms, as confirmed by the ¹H NMR analysis, and after the addition of the

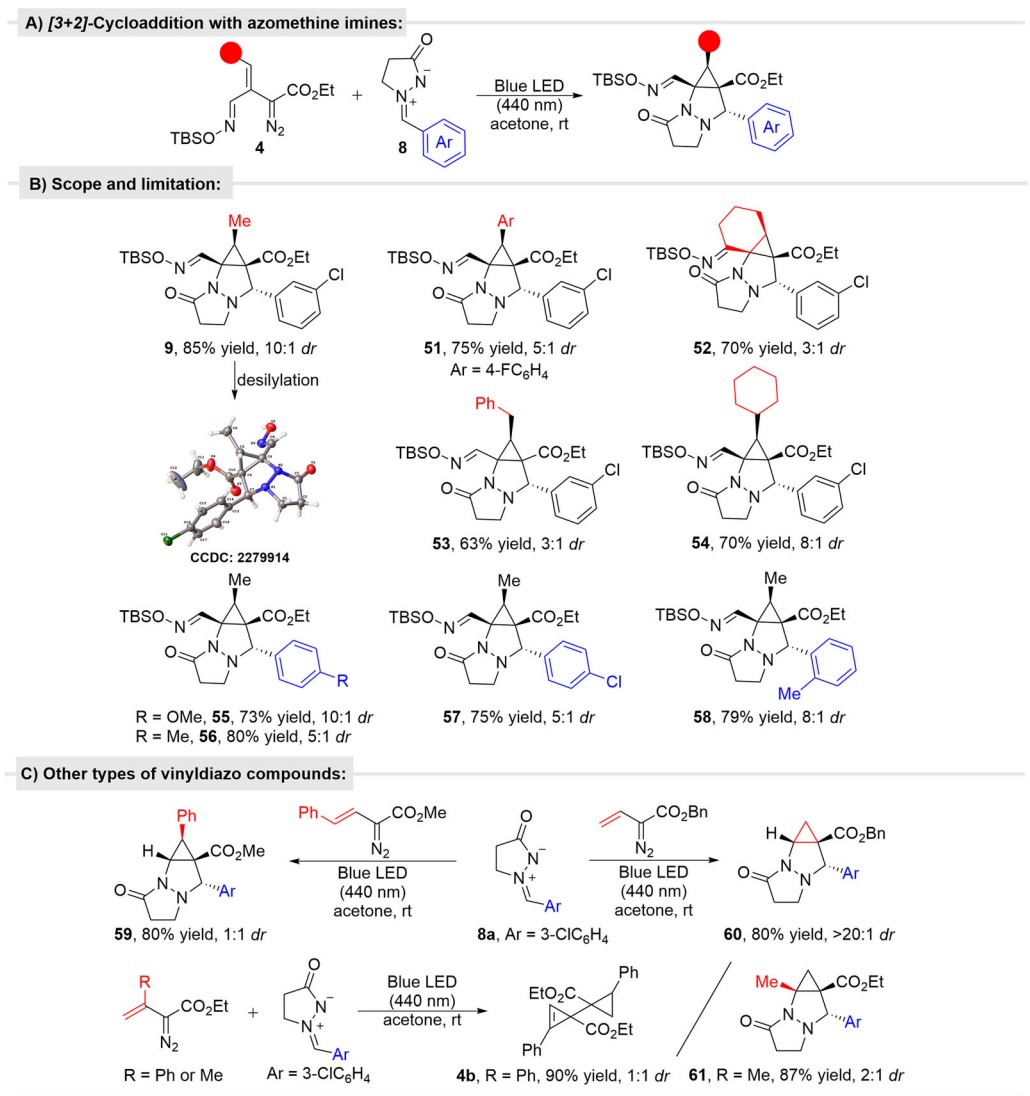

**Fig. 6 | Photocatalytic formal [3 + 2]-cycloaddition of diverse oximidovinyldiazo esters with azomethine imines. A** The general transformation; **B** Reaction scope of oximidovinyldiazoacetates and azomethine imines; **C** Reaction of other types of vinyldiazo compounds with azomethine imines.

nitrone, thermal cycloaddition occurs. Because cyclopropene can undergo rapid dimerization via an ene reaction[40], intermediate **3b** was not isolated in a pure form. According to DFT calculations, cycloaddition of nitrone **6** to the model cyclopropene (TMS protected) proceeds with a Gibbs free energy of activation of 85.2 and 90.0 kJ/mol for major and minor isomers, respectively, which matches the observed diastereoselectivity for the reaction with cyclopropene **3b**. Expectedly, for the reaction of simple acrylate (not a strained analog of cyclopropene) with nitrone **6** a barrier of 109.1 kJ/mol was calculated, corroborating the higher reactivity of strained cyclopropene over acrylate.

To test the generality of the developed tandem transformation, we also explored azomethine imines and nitrile oxides as other dipole-type substrates. Both reactions gave [3 + 2] products; for azomethine ylide **8** cycloadduct **9** was observed, while an unexpected pyridine *N*-oxide **11** formed from the cycloadduct produced in reactions of some oximidovinyldiazo esters with 2,4,6-trimethylbenzonitrile oxide (**10**) (Fig. 4).

Realizing the generality of the proposed strategy and the importance of heterocyclic scaffolds, we next optimized the reaction conditions and evaluated the substrate scope for the newly developed [2 + 3]-cycloaddition reactions.

**Cycloaddition of oximidovinyldiazo acetates with nitrones.** The model reaction of oximidovinyldiazo acetate **5a** with diphenylmethanimine oxide (**6**) was performed in different solvents, and the highest *dr* ratio (7:1) and yield (91% isolated) was obtained in acetone (see Table S1 in SI). Reactions in chlorocarbon solvents (dichloromethane, chloroform, 1,2-dichloroethane) also occurred in high yields, but their *dr* were significantly diminished to only 1:2 to 1:3. Using LED light sources (40 W) at 400 nm produced the cycloaddition product **7** in only 30% yield. The dominant stereoisomer in all cases is the one in which the phenyl group is *trans* to the carboethoxy group. The stereochemistry of the two diastereomers was determined spectroscopically based on the X-ray structure of the dominant isomer **30** after desilylation (Fig. 5).

The scope of the reaction was determined with a broad spectrum of oximidovinyldiazo compounds **5** and a representative selection of nitrones **6** (Fig. 5).

Reactions with alkyl and aryl substituted oximidovinyldiazo compounds **5** under the optimal conditions gave isooxazolidine products **7, 12–24** in high yields and modest *dr* values (Fig. 5B). Steric factors appear to play a major role in determining diastereoselectivity. The highest *dr* was achieved with the vinyldiazo ester without a

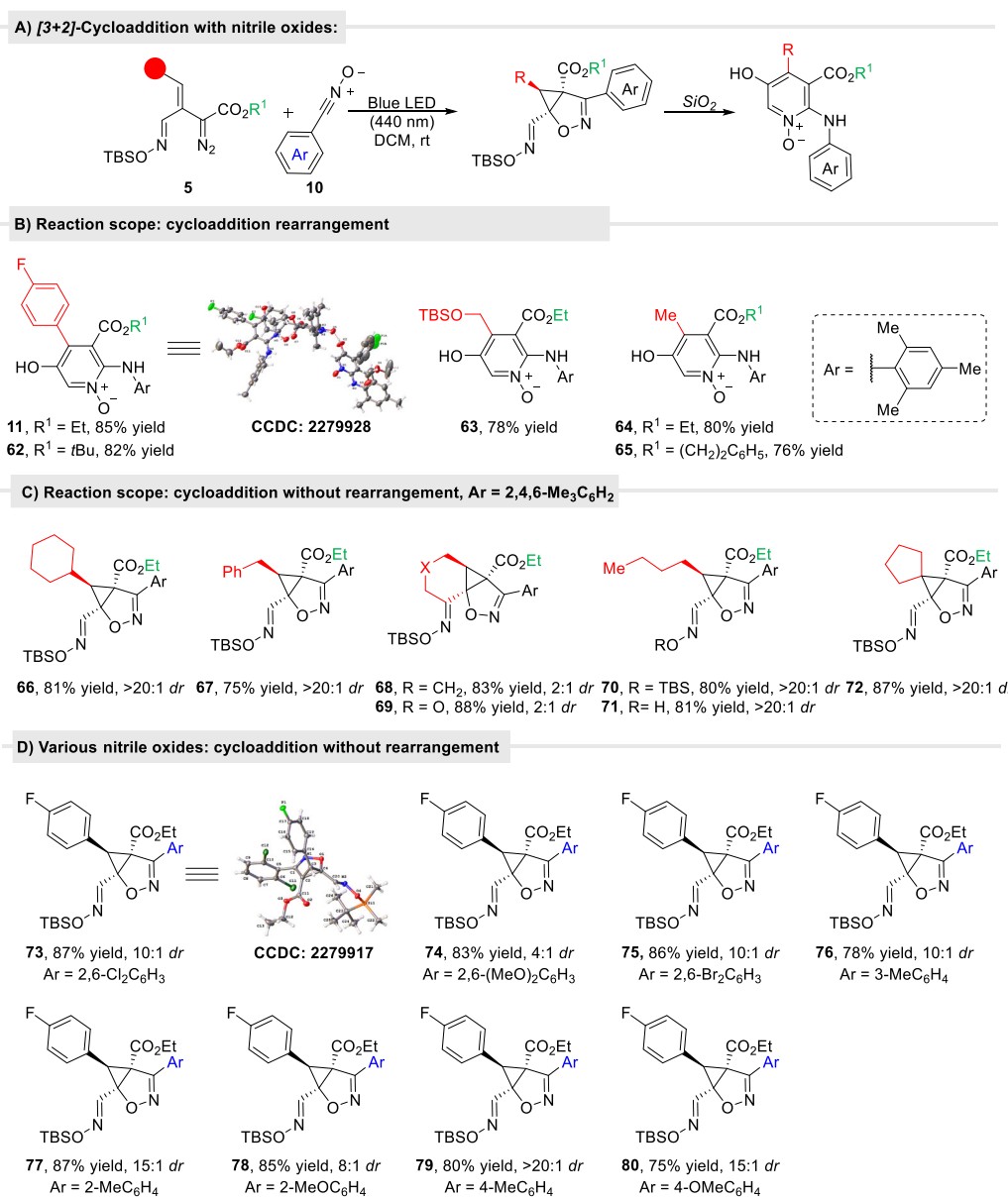

**Fig. 7 | Photocatalytic formal [3 + 2]-cycloaddition of oximidovinyldiazo esters with nitrile oxide compounds. A** The general transformation; **B** Formation of pyridine-*N*-oxides from photolytic cyclization/silica induced rearrangement. X-ray structure of **56** is of the hydrogen-bonded dimer; **C** Reaction scope for [3 + 2]-cycloaddition of oximidovinyldiazoacetates **5** with mesitylnitrile oxide (**10**); **D** Reaction scope of oximidovinyldiazo acetate **5a** with various arylnitrile oxides.

substituent at the gamma position (**22**), and the lowest *dr* intriguingly comes from the reaction of the vinyldiazo ester with a bulky menthyl ester of oximidovinyldiazoacetate **24**. In all these reactions, the competing ene reaction of intermediate cyclopropene **3b** was only a minor component of the reaction products, amounting to less than 10% yield; the only exception being the reaction of vinyldiazoester with no substituent at the gamma position leading to product **22**.

Similarly, the size and the nature of substituents on the nitrone influence the diastereoselectivity of the reaction (Fig. 5C). For Ar and alkyl substituted nitrones, yields remained at the same level (**25–33**) while for glyoxalic acid derived nitrone reactions were less efficient (**34, 36**). The yields, however, improved when deprotected oximidovinyldiazoacetate was used as a starting material, suggesting that the free hydroxyl group, by forming hydrogen bonds influences the reaction's efficacy (products **35, 37**). The biggest impact on the stereoselectivity of the reaction had the replacement of *N*-Bn with bulky *t*Bu group for which reactions produced only one diastereoisomer (compare **34** with **36**).

Oximidovinyldiazo compounds **5** are stable reagents in contrast to vinyldiazo and arylvinyldiazo derivatives. However, when tested in the developed method, gratifyingly, desired cycloaddition products **38–47** were also formed from these less stable derivatives, though with slightly diminished yields compared to reactions with oximidovinyldiazoacetates (Fig. 5D). The outcome of the photolytic reaction with aryl-substituted vinyldiazoacetates show contrasting behavior; with β- phenylvinyldiazoacetate the yields and *dr* of isoxazolidine cycloaddition products **38–40** were comparable to those with similarly substituted oximidovinyldiazoacetates, whereas with γ- phenylvinyldiazoacetate (styryldiazoacetate) competition with intramolecular pyrazole **2a** formation reduced the yield of the isooxazolidine cycloaddition product **41** which was formed with high diastereocontrol. In general, reactions of vinyldiazo compounds with nitrones derived from glyoxylic acid ester were less efficient. Though diastereoselectivity remained at the same level for *N*-Bn derivatives, *N*- *t*Bu analogs proved superior in furnishing products **42** and **44** as single diastereoisomers.

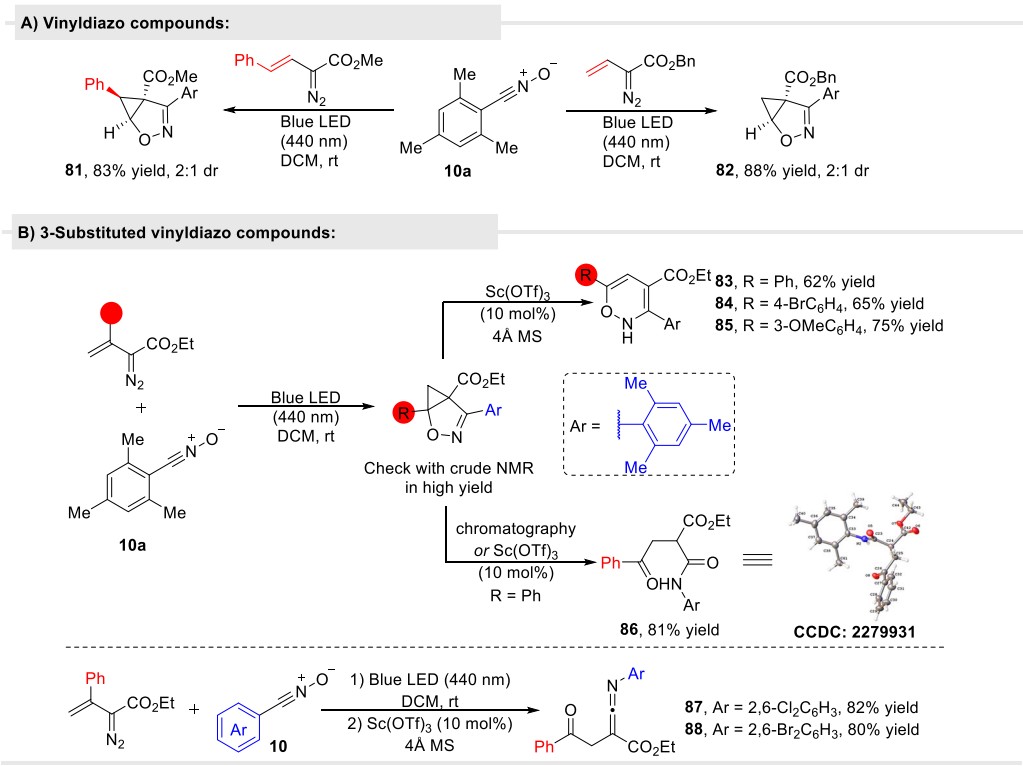

**Fig. 8 | Reaction of other vinyl diazo compounds with nitrile oxide. A** Cycloaddition of other types of vinyldiazo compounds with arylnitrile oxides. **B** Photocatalytic formal [3 + 2]-cycloaddition of 3-substituted vinyldiazo compounds with nitrile oxide compounds.

Furthermore, preliminary experiments confirmed that silyl-protected enol diazoacetate **48**, which forms a stable donor-acceptor cyclopropene, is also a suitable substrate in the developed tandem transformation (Fig. 5E). In this case, the bicyclic product was, however, not isolated. During the purification, the TIPS protecting group cleaves inducing the subsequent rearrangement giving product **50** in 39% yield.

**Cycloaddition of oximidovinyldiazo acetates with *N,N*-cyclic azamethine imines.** *N,N*-Cyclic azamethine imines are also suitable dipoles in cycloaddition reactions that have been reported to undergo [3 + 2]-cycloaddition reactions with propargylic and α,β-unsaturated carbonyl compounds[50,51], and cyclic enamines[52]. With vinyldiazo compounds, these azomethine imines are unreactive unless a transition metal catalyst converts the vinyldiazo compound to a metallovinylcarbene which then undergoes a N-N cleavage reaction to form a diimide with azomethine rather than [3 + 3]-cycloaddition[53,54]. In our case, photochemical tandem reaction of *N,N*-cyclic azamethine ylides **8** with vinyldiazo compound **5a** gave tricyclic pyrazolone **9** without the use of any catalyst (Fig. 6B).

Yields of cycloaddition reactions are comparable to those obtained for nitrones without the need for fine tuning of the reaction conditions. The competing ene dimer **4b** was obtained at most in less than 10% yield. Diastereoselectivities are also on a similar level with the *trans*-R/Ar isomer dominant (configuration certified by X-ray analysis). Interestingly, the reaction with styryldiazoacetate gave the [3 + 2]-cycloaddition product **59** in high yield but with no diastereoselectivity and without the evidence of pyrrole formation (Fig. 6C). In contrast, only one diastereoisomer **60** is produced from the reaction with the unsubstituted ethyl 2-diazo-3-butenoate. With ethyl 3-phenyl-2-diazo-3-butenoate, only the ene dimer **4a** was isolated, and there was no evidence for the formation of the [3 + 2]-cycloaddition product, whereas reaction with ethyl 3-methyl-2-diazo-3-butenoate resulted in a

high yield of the cycloaddition product **61** although with low diastereocontrol (Fig. 6C).

**Cycloaddition of oximidovinyldiazo acetates with nitrile oxides.** Nitrile oxides are relatively reactive dipolar species that are well known to undergo [3 + 2]-cycloaddition reactions with alkenes and alkynes to form diverse heterocyclic compounds[55–57], that have been used in the synthesis of natural products[58]. We selected the relatively stable aromatic nitrile oxide **10a** with the mesityl aromatic ring that does not dimerize and combined this nitrile oxide with vinyldiazoesters. As anticipated, under photolysis with blue light, in the presence of nitrile oxide **10a**, cyclopropene intermediate **3a** undergoes rapid [3 + 2]-cycloaddition with high diastereoselectivity (Fig. 7).

Intriguingly, some of the cycloaddition products converted to new chemical structures (**11, 62–65**) during chromatography on silica gel, of which one was identified both spectroscopically and by X-ray crystallography to be pyridine-*N*-oxides **11** (Fig. 7B). However, [3 + 2]-cycloaddition products **66–72** did not undergo rearrangement on silica gel even over long periods of time, or in the presence of Lewis acids (Fig. 7C). Products **73–80**, obtained from other aryl nitrile oxides in high yield, were stable and, intriguingly, did not rearrange regardless of the position of substituents and the electronic character of the phenyl ring of nitrile oxide (Fig. 7D). Reactions with vinyldiazo esters other than the oximidovinyldiazo esters **5** also formed stable cycloaddition products (**81** and, **82**) but with low stereoselectivity (Fig. 8A). As acidic conditions seemed to promote conversion of isoxazole products to pyridine-*N*-oxides, we wondered whether Sc(OTf)₃ treatment of the products from [3 + 2]-cycloaddition of 3-substituted vinyl diazo esters with mesitylnitrile oxide (**10a**) would lead to pyridine derivatives (Fig. 8B). In this case however, the reaction resulted in ring opening of the cyclopropane ring and formation of 2*H*-1,2-oxazine **83–85** in the absence of water or afforded cyclic product **86** in the presence of water. On the other hand, ketenimine products (**87** and

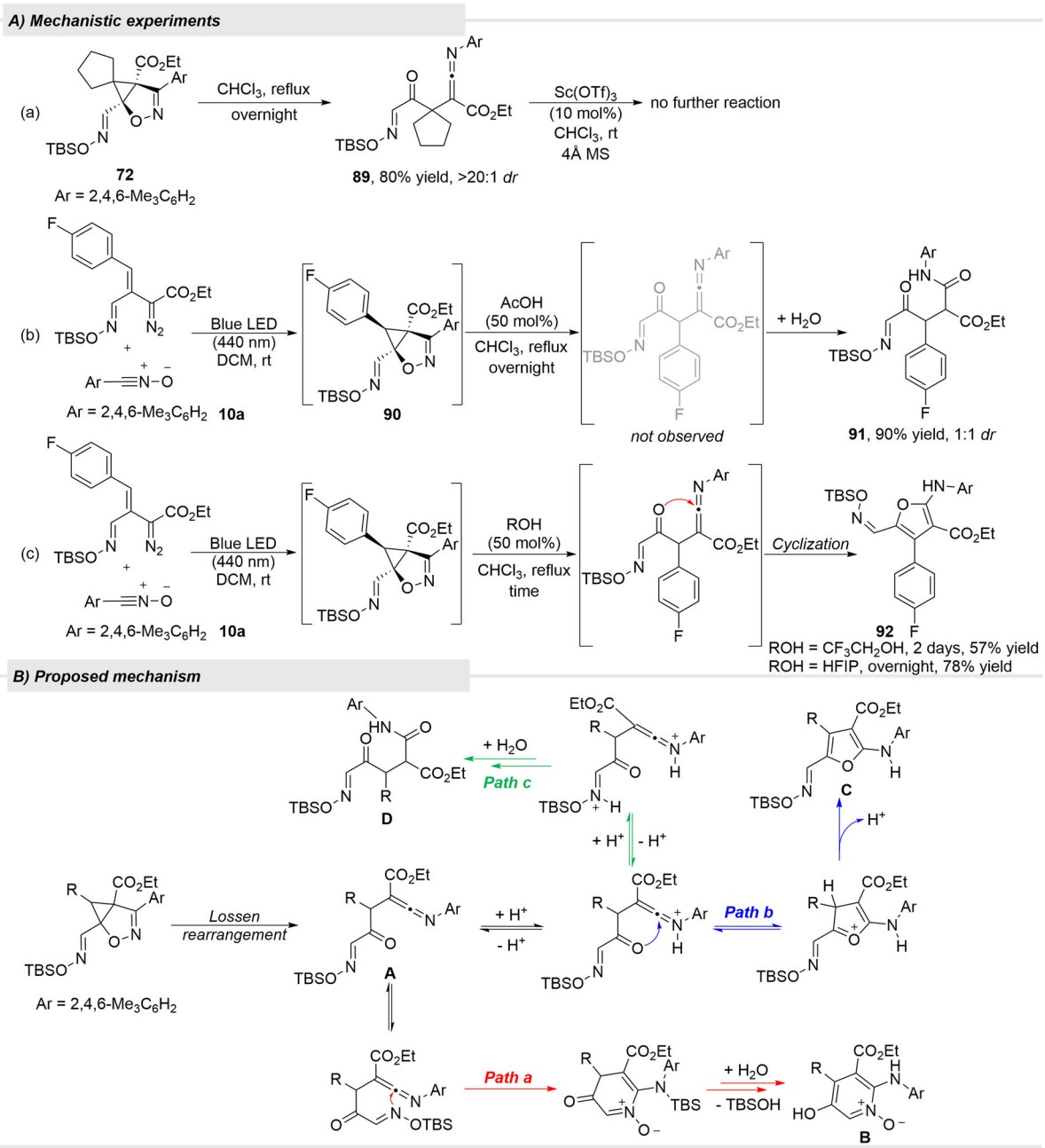

**Fig. 9 | Mechanistic studies for rearrangement of the [3 + 2]-cycloaddition product from ketenime intermediate to pyridine-*N*-oxide, furan, and ester/amide products. A** Control experiments. **B** Proposed mechanism.

88) were obtained in the two-step reaction from ethyl 2-diazo-3-phenylbut-3-enoate with arylnitrile oxides (Fig. 8B).

To determine how the pyridine-*N*-oxide was formed, we subjected several of the [3 + 2]-cycloaddition products to heating at reflux in chloroform. Only spirocyclopentyl derivative **72** underwent reaction and the product of this thermal reaction was ketenimine **89**, anticipated to be formed by the Lossen rearrangement, which was previously reported to occur in a dirhodium(II)-catalyzed process involving formation of an intermediate [3 + 2]-cycloaddition product similar to **89** (Fig. 9A, a)[59]. Indeed, when cycloaddition product **90** from cycloaddition of 4-fluorophenyl oximidovinyldiazoacetates with mesitylnitrile oxide **10a** was treated with AcOH in chloroform, the mixed ester/amide **91** as formed in 90% yield, which is similar to the product from the dirhodium(II) catalyzed reaction, thus also pointing to a ketenimine intermediate (Fig. 9A, b). The same treatment of the

benzyl and cyclohexyl analogs of **68** produced the corresponding ester/amides in 77% (2:1 *dr*) and 83% (3:1 *dr*) yields, respectively. Consistent with this intermediate, treatment of **90** with the less acidic 2,2,2-trifluoroethanol or hexafluoroisopropyl alcohol (HFIP) formed furan **92** in 57% and 78% yield, respectively, that further confirms the ketenimine intermediate (Fig. 9A, c).

The formation of ketenimines intermediates from the isoxazolerelated products obtained from nitrile oxides explains the formation of pyridine-*N*-oxides, ester/amides, and furan products. In the proposed mechanism, ketenimine **A** is formed by the Lossen rearrangement[60,61], although the reactivity of the [3 + 2]-cycloaddition products towards this rearrangement is not evident. This multisubstituted intermediate has two basic centers (the imine and oxime nitrogens) whose protonation influences subsequent reactions (Fig. 9B). Although protonation may not be required for the formation

of the pyridine-*N*-oxide **B** (path a), protonation of the imine nitrogen of intermediate, **A** activates the central ketenimine carbon for nucleophilic attack by the ketone oxygen to form furan derivative **C** (path b). If both basic nitrogens are protonated, neither intramolecular reaction occurs, and ketenimine hydrolysis occurs to produce **D** (path c).

## Conclusions

In conclusion, we have found that direct photolysis of vinyldiazo compounds selectively leads to cyclopropenes, which contrasts with the same reactions under thermal conditions that favors the formation of pyrazoles. Once formed, these reactive cyclopropene intermediates undergo [3 + 2]-cycloaddition with diverse dipolar species to yield heterocyclic scaffolds, highly prized by the pharmaceutical industry. Reactions of vinyldiazo compounds with nitrones, *N,N*-cyclic azamethine ylides, and nitrile oxides afforded heterocyclic products in high yields with mostly good stereoselectivities. The most diastereoselective reactions with nitrones were those derived from *N-t*Buglyoxylic acid, for which single isomers were observed. *N,N*-cyclic azamethine ylides react with the photochemically-generated cyclopropenes in modest to good yields and diastereoselectivities, and the transformation is general for vinyldiazoacetates. Nitrile oxides show similar generality in its cycloaddition reactions with photochemically-generated cyclopropenes, but its bicyclic isoxazole products exhibit a diversity of reactions and reactivities, some of which, especially the formation of pyridine-*N*-oxides, were unexpected. A great variety of heterocyclic scaffolds accessible by the developed method emphasizes the importance of cyclopropene generated from diazo compounds in a photochemical manner. Further studies of its unique reactivity are currently underway in our laboratory.

## Methods

### General procedure for [3 + 2]-cycloadditions with nitrones

To a 10-mL oven-dried vial with a magnetic stirring bar, vinyldiazo compound (0.1 mmol) in 1.0 mL acetone was added over 1 min to a solution of nitrones (0.12 mmol, 1.2 equiv.) in the same solvent (1.0 mL) at room temperature with irradiation by 440 nm blue LED (40 W), and the reaction mixture was stirred for 5–15 min under these conditions. When the reaction was complete (monitored by TLC), the reaction mixture was purified by flash column chromatography on silica gel without additional treatment (hexanes: EtOAc = 20:1 to 15:1) to give the pure [3 + 2]-cycloaddition products in good yields.

### General procedure for [3 + 2]-cycloaddition with azamethine imines

To a 10-mL oven-dried vial with a magnetic stirring bar, vinyldiazo compound (0.12 mmol, 1.2 equiv.) in 1.0 mL acetone was added to a solution of azamethine imine (0.1 mmol) in the same solvent (1.0 mL) via a syringe pump over 2 h at room temperature with irradiation by 440 nm blue LED (40 W). When the reaction was complete (monitored by TLC), the reaction mixture was purified by flash column chromatography on silica gel without additional treatment (hexanes: EtOAc = 20:1 to 15:1) to give the pure [3 + 2]-cycloaddition products in good yields.

### General procedure for [3 + 2]-cycloaddition with nitrile oxides

To a 10-mL oven-dried vial with a magnetic stirring bar, vinyldiazo compound (0.12 mmol, 1.2 equiv.) in 1.0 mL DCM was added to a solution of the nitrile oxide (0.1 mmol) in the same solvent (1.0 mL) via a syringe pump over 1 h at room temperature with irradiation by 440 nm blue LED. When the reaction was complete (monitored by TLC), the reaction mixture was purified by flash column chromatography on silica gel without additional treatment (hexanes: EtOAc = 20:1 to 1:1) to give the pure [3 + 2]-cycloaddition and cycloaddition/rearrangement products in good yields.

## Data availability

The Authors declare that all relevant data generated and analyzed during this study, which include experimental, spectroscopic, crystallographic and computational data, are included in this article and its supplementary information. Should any raw data files be needed in another format, they are available from the corresponding author upon request. Crystallographic data for the structures reported in this article have been deposited at the Cambridge Crystallographic Data Center under deposition numbers CCDC 1) 2282181, 2) 2295973, 3) 2279914, 4) 2279928, 5) 2279917, 6) 2279931. Coordinates of the optimized structures are present as source data. Copies of the data can be obtained free of charge via https://www.ccdc.cam.ac.uk/structures/. Source data are provided in this paper.

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

## Acknowledgements

Financial support for this work was supported by the U.S. National Science Foundation (CHE 2054845) for M.P.D. and M. B. and by NCN Poland OPUS UMO-2019/35/B/ST4/03435 for D.G.and K.Ł. Calculations have been carried out using resources provided by Wrocław Center for Networking and Supercomputing (http://wcss.pl), grant No. 518. for W.CH.

## Author contributions

M. B. and K. Ł. performed photo-catalytic studies of the vinyldiazo compounds and characterization of products. M. B. and M. Baird prepared the vinyl diazo compounds. M. B., D. G., and M. P. D. conceived and designed the experiments, W. CH. prepared DFT calculations, and M. B., K. Ł., D. G., and M. P. D. prepared the manuscript. All authors contributed to discussions and commented on the manuscript.

## Competing interests

The authors declare no competing interests.
