## [Peer Review File · Nature Communications]

Photo-cycloaddition reactions of vinyl diazo compoundsREVIEWER COMMENTS

Reviewer #1 (Remarks to the Author):

Ming Bao et.al reported a photocycloaddition of vinyldiazos. The methodology is featured by broad substrate scope, good to excellent stereoselectivity and chemoselectivity. Evolving the vinyldiazo-involved cycloaddition from the meta-catalysed version to the metal free version is of pivotal significance in amplifying the scope of reactivity and in the application in the drug discovery. Overall, this paper is strongly recommended to be published in Nature Communications after minor revision. The following are the tips for revision:

1. Line68-75 is suggested to move to Supplementary Material as the protocol of the DFT calculations.
2. Line 109, intermediate 3b or 3a? cause no 3b was found in the fig. 3A.
3. Line112-114, a figure for the reactivity comparison via DFT-calculations should be added in the Supplementary Material.
4. How about asymmetric catalysis? The publications including *Org. Lett.* 2019, 21, 1, 40–44; *J. Am. Chem. Soc.* 2017, 139, 46, 16506–16509 represent good examples for the enantioselective cycloaddition of cyclopropenes.
5. The publications about photo-catalyzed decomposition of diazocompounds are suggested to cite: Blue light-promoted photolysis of aryldiazoacetates. *Chem. Sci.*,2018,9, 5112-5118; Photo-Induced Homologation of Carbonyl Compounds for Iterative Syntheses, *Angew. Chem.Int. Ed.*2022,61, e20221157.
6. Page 5, Under light irradiation cyclopropene forms as confirmed by the ¹H NMR analysis and after the addition of nitrene the thermal cycloaddition occurs.", the " nitrene " should be "nitron".

Reviewer #2 (Remarks to the Author):

In this manuscript, Doyle and co-workers reported a photochemical tandem reaction for produce diverse heterocyclic scaffolds from vinyl diazo compounds: via the direct photolysis of vinyl diazo compounds leads to cyclopropenes intermediates, and then undergo [3+2]-cycloaddition with diverse dipolar species (include nitrones, N,N-cyclic azamethine ylides, and nitrile oxides) to yield the heterocyclic scaffolds in high yields with good stereoselectivities. It's an interesting work, especially in the parts of cycloaddition of oximidovinyl diazo acetates with nitrile oxides (Fig 7). What's more, experiment results are supported by their computational studies; and the further control experiments were carried out to account for the reaction mechanism. Last, this work is also a meaningful extension of the author group's previous work.

The SI seems of high quality. Compounds are fully characterized. The paper was well-written.

I recommend publication of this work in Nature Communications after revision.

- 1) Silyl-protected enol diazoacetates are known to form stable donor-acceptor cyclopropenes, does it could be a suitable substrate in this tandem method?
- 2) SI, part 3-6 should provide a title reaction graphics, respectively.
- 3) The author should give more detailed description about the Figure 8, in main text.

Reviewer #3 (Remarks to the Author):

This manuscript by Ming Bao presents their computational and experimental studies to reveal that under light irradiation these diazo reagents selectively transform into cyclopropenes which engage in cycloaddition reactions with various dipoles. Although the selected system is worthy of in-deepen research, the detailed calculation and comparison are lacking. Thus, I do not recommend its publication in nature communications with this current manuscript. The detailed comments describe as below:

1. In the described section of theoretical method "Structures of minima and transition states were optimized employing B3LYP and configuration interaction singles (CIS) with 6-31G(d) basis set for ground (S0) and excited states (S1), respectively." of the manuscript and

ESI. However, the detailed information of Gibbs free energy in Figure 2B, Figure 3 and Table (of ESI) is not provided.

2. In the ground (S_0), the model vinyl diazoacetate I towards pyrazole II is preferred with $\Delta G^\ddagger = 115.6$ kJ/mol. In the excited states (S_1), Cycloisomerization of methyl 2-diazobut-3-enoate V to pyrazole system VII features considerable barrier ($\Delta G^\ddagger = 118.1$ kJ/mol). Compared them, the difference is only 0.597 kcal/mol (2.5 kJ/mol) which is very small. The error from the computational methods or model is considered. Thus, the high level method or other complex model should be used.

3. In the description of "Thus, it presumably first undergoes internal conversion to ground state III, at which it then converts into cyclopropene system IV.", the more convincing evidence should be considered.

4. "Because cyclopropene can undergo rapid dimerization via an ene reaction,²² intermediate 3b was not isolated in a pure form. According to DFT calculations cycloaddition of nitrene 6 to cyclopropene 3b proceeds with a Gibbs free energy of activation of 85.2 and 90.0 kJ/mol for major and minor isomers, respectively, which matches the observed diastereoselectivity. Expectedly, for the reaction of simple acrylate (not strained analogue of cyclopropene) with nitrene 6 a barrier of 109.1 kJ/mol was calculated, corroborating the higher reactivity of strained cyclopropene over acrylate." However, in Figure 3, the information on intermediate 3b is not provided.

Manuscript ID: **Nature Communications manuscript NCOMMS-23-49242-T**

Title: *Photo-cycloaddition reactions of vinyl diazo compounds*

Authors: Ming Bao, Klaudia Łuczak, Wojciech Chaładaj, Marriah Baird, Dorota Gryko, Michael P. Doyle

RESPONSE TO REVIEWERS

We thank the referees for their kind and meaningful remarks regarding our submission, which helped us improve our work, and for suggesting its publication in *Nature Communications*. We have done our best to address all concerns according to the suggestions of the reviewers. We have also resolved all technical issues as requested. A detailed description of all the changes made is listed below. Additionally, a version of the manuscript with all changes highlighted by giving the text a yellow background is uploaded.

REVIEWER 1:

Comments:

1. *Line 68-75 is suggested to move to Supplementary Material as the protocol of the DFT calculations.*
} The detailed description of the DFT methodology was moved to Supplementary material.

2. *Line 109, intermediate 3b or 3a? cause no 3b was found in the fig. 3A.*
In Fig. 3A the typo was corrected, the cyclopropene should have number **3b**, not **3a**, and now it is assigned as **3b**.

3. *Line 112-114, a figure for the reactivity comparison via DFT-calculations should be added in the Supplementary Material.*
} A suitable figure has been added to the Supplementary material in Section 9, page 151.

4. *How about asymmetric catalysis? The publications including *Org. Lett.* 2019, 21, 1, 40–44; *J. Am. Chem. Soc.* 2017, 139, 46, 16506–16509 represent good examples for the enantioselective cycloaddition of cyclopropenes.*

Indeed, asymmetric catalysis is a continual goal of our research. However, the approaches mentioned by this reviewer take the reactions onto a different mechanistic pathway. The *Organic Letters* article requires the formation of a metal carbene intermediate in which the chiral ligand of the metal bound to the carbene imparts asymmetric induction. Unlike enoldiazoacetates, however, the vinyl diazoacetates described in our manuscript do not form metalvinylcarbenes upon treatment with metal catalysts (see reference 23). Similarly, the *J. Am. Chem. Soc.* article describes a process wherein the chiral lanthanum catalyst undergoes addition to the cyclopropene double bond at the same time as the nucleophile is captured, and then the C=C bond of the nucleophile coordinates to the lanthanum to facilitate subsequent product formation – a process clearly unrelated to the cycloaddition processes described in our manuscript. There is a possibility that organocatalysis may provide an opening to enantioselective transformations, but this has not been attempted by us or achieved by others.

5. *The publications about photo-catalyzed decomposition of diazocompounds are suggested to cite: *Blue light-promoted photolysis of aryldiazoacetates. Chem. Sci.*, 2018, 9, 5112-5118; *Photo-Induced Homologation of Carbonyl Compounds for Iterative Syntheses, Angew. Chem. Int. Ed.* 2022, 61, e20221157.*

These works as well as others on photochemical transformations have been added as refs. 1d-g

6. Page 5, Under light irradiation cyclopropene forms, as confirmed by the ¹H NMR analysis and after the addition of nitrene the thermal cycloaddition occurs.", the " nitrene " should be "nitrene".

Corrected

REVIEWER 2:

Comments:

1. Silyl-protected enol diazoacetates are known to form stable donor-acceptor cyclopropenes, does it could be a suitable substrate in this tandem method?

The authors thank the reviewer for this meaningful question. Silyl-protected enol diazoacetates form donor-acceptor cyclopropenes that are, indeed, a suitable substrate in our tandem method. However, in this case the bicyclic product forms, but during the purification process, the TIPS protecting group is cleaved and rearrangement occurs, leading to a six-membered ring (39%). This example is included in Figure 5 and the product is assigned as **48**. The adequate description is also given in the text.

2. SI, part 3-6 should provide a title reaction graphics, respectively.

Reaction graphics are added.

3. The author should give more detailed description about the Figure 8, in main text.

We have given a more precise description of Figure 8, and we hope that it is clear now.

REVIEWER 3:

Comments:

1. In the described section of theoretical method "Structures of minima and transition states were optimized employing B3LYP and configuration interaction singles (CIS) with 6-31G(d) basis set for ground (S0) and excited states (S1), respectively." of the manuscript and ESI. However, the detailed information of Gibbs free energy in Figure 2B, Figure 3 and Table (of ESI) is not provided.

The description of the DFT methodology used was moved to Supplementary Material as requested by Reviewer 1. To avoid ambiguity, this section was revised, and detailed information on the calculation of the Gibbs free energy was added.

2. In the ground (S0), the model vinyl diazoacetate I towards pyrazole II is preferred with $\Delta G^\ddagger = 115.6$ kJ/mol. In the excited states (S1), Cycloisomerization of methyl 2-diazobut-3-enoate V to pyrazole system VII features considerable barrier ($\Delta G^\ddagger = 118.1$ kJ/mol). Compared them, the difference is only 0.597 kcal/mol (2.5 kJ/mol) which is very small. The error from the computational methods or model is considered. Thus, the high level method or other complex model should be used.

We would like to point out that the presented two pathways are considered in the excited state, of which extrusion of nitrogen is practically barrierless at the considered level of theory, while cyclization is associated with a barrier of more than 100 kJ/mol (**TS5** vs. **TS6**, respectively). Therefore, the formation of carbene is strongly preferred. A comparison of transition states in ground (TS1 and TS2) and excited states (TS5 and TS6) is not valid here, and this approach is not relevant to understanding of the developed process as it takes place under light irradiation (e.g. in the excited state). The calculations for the ground state have been included to only illustrate differences in the reactivity of diazo compounds under thermal (ground state) and photochemical (excited state) conditions.

According to our calculations, under ambient temperature no reaction of the diazo compound in the ground state (S0) is expected (the diazo compound is stable at ambient temperature). It is consistent with the relatively high barrier calculated for both thermal processes (**TS1** and **TS2**, $\Delta G^\ddagger > 100$ kJ/mol) and is in accordance with the experimental stability observed of the starting material at room temperature. Under light irradiation, if a facile pathway is accessible in the excited state (**TS5**, $\Delta G^\ddagger = 3.4$ kJ/mol, extrusion of N₂ in the investigated

system) and is preferred over the considerably higher barrier (at S1) of a side reaction (cyclization in the excited state, **TS6**, $\Delta G^\ddagger = 118.1$ kJ/mol). ΔG^\ddagger values and their relationship to the ground state barriers is not valid as our reactions take place under light irradiation.

Consequently, although high-level theoretical investigations and advanced spectroscopic studies are always beneficial for understanding of photophysical processes, we believe that the applied level of theory is sufficient to understand and rationalize the chemistry of methyl 2-diazobut-3-enoate. We have, however, performed a further theoretical investigation to provide a more complete picture of the investigated system: a) dimerization of cyclopropene **VI** via the ene reaction (TS and the product) and b) chemistry of the carbene in the triplet state were added to scheme 2. Please, note that the introduced additions caused a change in the labelling of the structures in Figures 2 and 3.

- 3. In the description of "Thus, it presumably first undergoes internal conversion to ground state III, at which it then converts into cyclopropene system IV.", the more convincing evidence should be considered.*

The equilibrium geometry of carbene **IX** (**VI** in original paper) in the excited state is flat, contrasting with the bend geometry of the corresponding carbene in the ground state. Therefore, the ground-state energy for the flat conformer (minimum at S1) is considerably higher than for the bend one and close in its energy in the S1 state. The small energy gap between S0 and S1 for the flat geometry renders radiationless decay plausible followed by relaxation of the system. A short section with details was added to SI (section 9).

- 4. Because cyclopropene can undergo rapid dimerization via an ene reaction,²² intermediate 3b was not isolated in a pure form. According to DFT calculations cycloaddition of nitrene 6 to cyclopropene 3b proceeds with a Gibbs free energy of activation of 85.2 and 90.0 kJ/mol for major and minor isomers, respectively, which matches the observed diastereoselectivity. Expectedly, for the reaction of simple acrylate (not strained analogue of cyclopropene) with nitrene 6 a barrier of 109.1 kJ/mol was calculated, corroborating the higher reactivity of strained cyclopropene over acrylate. " However, in Figure 3, the information on intermediate 3b is not provided.*

The intermediate in Scheme 3A was wrongly labelled **3a** (should be **3b**). It was corrected. We apologize for the confusion our mistake caused.

REVIEWER COMMENTS

Reviewer #3 (Remarks to the Author):

Some information were revised and provided. However, some errors still exist in this current manuscript. Thus, this work could be published in Nature Communications after detailed revision. The detailed comments describe as below:

1. In the current manuscript, the description is confused and some errors present in Figure 2B, such as, two labels "TS5", the value of TS4 is "-60.6" or "60.6".

Such as, "Dimerization of IV via ene-reaction require to overcome barrier of $\Delta G^\ddagger = 110.8$ kJ/mol." However, the values of IV and TS4 are -50.2 and -60.6 kJ/mol respectively in Figure 2B.

"Similar reactivity pattern was observed for vinyl diazo compounds substituted with simple alkyl and aryl groups (see, SI)." This description was unclear. In SI, this content should be labeled in Table SX. In addition, the values ΔG^\ddagger in Table should be explained in detailed. Otherwise, it may be treated as thermodynamic energy due to the title of "Thermal reactivity of vinyl diazo compounds". In addition, in this Table, an example of "R1=H, R2=CNOTMS" is significantly different from other compounds. For this case, how to consider this anomalous result.

"Cycloisomerization of methyl 2-diazobut-3-enoate V to pyrazole system VII not only features considerable barrier ($\Delta G^\ddagger = 118.1$ kJ/mol) but is also highly endergonic ($\Delta G = 92.3$ kJ/mol)." "V" ought to "VIII", "VII" ought to "X".

Similar situation exist in "In contrast, excited vinyl diazoacetate V loses dinitrogen in a practically barrierless event ($\Delta G^\ddagger = 3.4$ kJ/mol). However, the subsequent cyclization of resulting carbene VI into cyclopropene ring cannot occur at S1 PES (high-energy cyclopropene in the excited state spontaneously opens towards vinyl carbene). Thus, it presumably first undergoes internal conversion to ground state III, at which it then converts into cyclopropene system IV. In contrast, cyclization of carbene VI in triplet state, which is a ground state, through TS5 is associated with very high barrier $\Delta G^\ddagger = 188.0$ kJ/mol, not accessible under reaction conditions."

2. In the comment of first revision, "In the description of "Thus, it presumably first

undergoes internal conversion to ground state III, at which it then converts into cyclopropene system IV.”, the more convincing evidence should be considered.”

Your response is “The equilibrium geometry of carbene IX (VI in original paper) in the excited state is flat, contrasting with the bend geometry of the corresponding carbene in the ground state. Therefore, the ground-state energy for the flat conformer (minimum at S1) is considerably higher than for the bend one and close in its energy in the S1 state. The small energy gap between S0 and S1 for the flat geometry renders radiationless decay plausible followed by relaxation of the system. A short section with details was added to SI (section 9).”

In section 9 of SI, some energies were indeed provided in Table. However, how does the geometries and the values of IX, III, VI in Figure 2B correspond to this table information. Thus, detailed information should be described.

3. In the described section of theoretical method “Structures of minima and transition states were optimized employing B3LYP and configuration interaction singles (CIS) with 6-31G(d) basis set for ground (S0) and excited states (S1), respectively. Frequency analysis was performed at the same level to provide correction to thermodynamic functions and confirm the nature of optimized structures (minima and transition states featured zero or one imaginary frequency, respectively).” “Gibbs free energies were calculated as a sum of electronic energy from single point calculations (involving solvation) and thermal correction to Gibbs free energy from frequency calculation.” of SI.

In the current manuscript, is the “thermal correction to Gibbs free energy from frequency calculation” obtained at B3LYP/6-31G(d) or CIS/6-31G(d) levels? The results from between B3LYP and CIS methods should be compared. Or, how to present the details of CIS method in theoretical process?

4. The font size is different. Such as “All the calculations were performed with Gaussian 16 package.¹¹ Structures of minima and transition states were optimized employing B3LYP and configuration interaction singles (CIS) with 6-31G(d) basis set for ground (S0) and excited states (S1), respectively. Frequency analysis was performed at the same level to provide correction to thermodynamic functions and confirm the nature of optimized structures

(minima and transition states featured zero or one imaginary frequency, respectively). Single point energies were calculated at M06/6-311+G(d,p) level of theory (time-dependent density functional theory was employed for structures in S1 excited states) employing solvation (acetone) with the SMD model.¹² Gibbs free energies were calculated as a sum of electronic energy from single point calculations (involving solvation) and thermal correction to Gibbs free energy from frequency calculation. Gibbs free energies were reported in kJ/mol in respect to starting materials (eg. in Figure 2 compound I in S0 state reported as 0 kJ/mol). Molecular structures were visualized in CYLview.” in SI.

Manuscript ID: **Nature Communications manuscript NCOMMS-23-49242-T**

Title: *Photo-cycloaddition reactions of vinyl diazo compounds*

Authors: Ming Bao, Klaudia Łuczak, Wojciech Chaładaj, Marriah Baird, Dorota Gryko, Michael P. Doyle

REVIEWER :

Comments:

1. *In the current manuscript, the description is confused and some errors present in Figure 2B, such as, two labels "TS5", the value of TS4 is "-60.6" or "60.6". Such as, "Dimerization of IV via ene-reaction require to overcome barrier of $\Delta G^\ddagger = 110.8$ kJ/mol." However, the values of IV and TS4 are -50.2 and -60.6 kJ/mol respectively in Figure 2B.*

We are grateful for catching this error. It was a typo and is now corrected, it should be 60.6. Now the value of ΔG^\ddagger is correct.

2. *Similar reactivity pattern was observed for vinyl diazo compounds substituted with simple alkyl and aryl groups (see, SI)." This description was unclear. In SI, this content should be labeled in Table SX. In addition, the values ΔG^\ddagger in Table should be explained in detailed. Otherwise, it may be treated as thermodynamic energy due to the title "Thermal reactivity of vinyl diazo compounds". In addition, in this Table, an example of "R1=H, R2=CNOTMS" is significantly different from other compounds. For this case, how to consider this anomalous result.*

The statement in the manuscript was revised to clarify and avoid ambiguity (page 3). The section concerning the comparison of thermal reactivity of various vinyl diazo compounds in SI was revised: a) graphic in the heading of Table S3 was corrected (carbene should be an intermediate in the upper reaction), b) the title of Table S4 was changed to clearly reflect that only computational results were compared. c) the title of the column was added, clearly stating that electronic energies are given there.

Furthermore, our experimental studies showed that heating of model compound **5a** in acetone at 50 °C does not give a cyclization product corroborating theoretical findings.

3. *Cycloisomerization of methyl 2-diazobut-3-enoate V to pyrazole system VII not only features considerable barrier ($\Delta G^\ddagger = 118.1$ kJ/mol) but is also highly endergonic ($\Delta G = 92.3$ kJ/mol)." "V" ought to "VIII", "VII" ought to "X".*

We apologise for not labelling the species correctly during the first revision process and for the confusion it caused. In fact, it should be **VIII** to **X** and **VII** to **X**. This has been corrected.

4. *Similar situation exist in "In contrast, excited vinyl diazoacetate V loses dinitrogen in a practically barrierless event ($\Delta G^\ddagger = 3.4$ kJ/mol). However, the subsequent cyclization of resulting carbene VI into cyclopropene ring cannot occur at S1 PES (high-energy cyclopropene in the excited state spontaneously opens towards vinyl carbene). Thus, it presumably first undergoes internal conversion to ground state III, at which it then converts into cyclopropene system IV. In contrast, cyclization of carbene VI in triplet state, which is a ground state, through TS5 is associated with very high barrier $\Delta G^\ddagger = 188.0$ kJ/mol, not accessible under reaction conditions."*

The same situation as described in point 3 occurred here. We apologise for not labelling the species correctly during the first revision process.

5. *In the comment of first revision, "In the description of "Thus, it presumably first undergoes internal conversion to ground state III, at which it then converts into cyclopropene system IV.", the more convincing evidence should be considered."Your response is "The equilibrium geometry of carbene IX (VI in original paper) in the excited state is flat, contrasting with the bend geometry of the corresponding carbene in the ground state. Therefore, the ground-state energy for the flat conformer (minimum at S1) is considerably higher than for the bend one and close in its energy in the S1 state. The small energy gap between S0 and S1 for the flat geometry renders radiationless decay plausible followed by relaxation of the system. A short section with details was added to SI (section 9)."In section 9 of SI, some*

energies were indeed provided in Table. However, how does the geometries and the values of IX, III, VI in Figure 2B correspond to this table information. Thus, detailed information should be described.

We have revised Section 9c that describes electronic energies of carbene **IX** in various conformations and electronic states.” In addition, we have also provided a relation between **IX**, **IXa**, **III**, and **VI** species. We hope that now it clearly explains the features of carbene **IX** and the description is more explicit.

- 6.** *In the described section of theoretical method “Structures of minima and transition states were optimized employing B3LYP and configuration interaction singles (CIS) with 6-31G(d) basis set for ground (S0) and exited states (S1), respectively. Frequency analysis was performed at the same level to provide correction to thermodynamic functions and confirm the nature of optimized structures (minima and transition states featured zero or one imaginary frequency, respectively).” “Gibbs free energies were calculated as a sum of electronic energy from single point calculations (involving solvation) and thermal correction to Gibbs free energy from frequency calculation.” of SI.*
In the current manuscript, is the “thermal correction to Gibbs free energy from frequency calculation” obtained at B3LYP/6-31G(d) or CIS/6-31G(d) levels? The results from between B3LYP and CIS methods should be compared. Or, how to present the details of CIS method in theoretical process?

Corrections to thermodynamic functions were taken from frequency calculations at the same level as the geometry optimizations. Thus, for a) structures in the ground state (S0) thermal corrections to the Gibbs free energy were taken from the frequency calculations at the B3LYP/6-31G(d) level of theory, while for b) structures in the exited state (S1) thermal correction to Gibbs free energy were taken from frequency calculations at the CIS/6-31G(d) level of theory.

Frequency calculations were performed at the same level as optimization to provide meaningful results. This ensures that the frequency calculations (second derivative of the energy with respect to the Cartesian nuclear coordinates) are computed exactly at a stationary point. Importance of this issue is stressed by authors of Gaussian 16 package in the description of a keyword freq: “Vibrational frequencies are computed by determining the second derivatives of the energy with respect to the Cartesian nuclear coordinates and then transforming to mass-weighted coordinates. *This transformation is only valid at a stationary point.* Thus, it is *meaningless* to compute frequencies at any geometry other than a stationary point for the method used for frequency determination. For example, computing 6-311G(d) frequencies at a 6-31G(d) optimized geometry produces meaningless results.”

For clarity, we have also revised the introduction part to calculation (in SI, section 9) and give more detailed information.

- 7.** *The font size is different. Such as “All the calculations were performed with Gaussian 16 package.¹¹ Structures of minima and transition states were optimized employing B3LYP and configuration interaction singles (CIS) with 6-31G(d) basis set for ground (S0) and exited states (S1), respectively. Frequency analysis was performed at the same level to provide correction to thermodynamic functions and confirm the nature of optimized structures (minima and transition states featured zero or one imaginary frequency, respectively). Single point energies were calculated at M06/6-311+G(d,p) level of theory (time-dependent density functional theory was employed for structures in S1 exited states) employing solvation (acetone) with the SMD model.¹² Gibbs free energies were calculated as a sum of electronic energy from single point calculations (involving solvation) and thermal correction to Gibbs free energy from frequency calculation. Gibbs free energies were reported in kJ/mol in respect to starting materials (eg. in Figure 2 compound I in S0 state reported as 0 kJ/mol). Molecular structures were visualized in CYLview.” in SI.*

It was corrected.

REVIEWERS' COMMENTS

Reviewer #3 (Remarks to the Author):

I agree with the current revision. This paper is recommended to be published in Nature Communications.